# Metabolic and Obesity Phenotype Trajectories in Taiwanese Medical Personnel

**DOI:** 10.3390/ijerph19138184

**Published:** 2022-07-04

**Authors:** Hsin-Yun Chang, Jer-Hao Chang, Yin-Fan Chang, Chih-Hsing Wu, Yi-Ching Yang

**Affiliations:** 1Institute of Allied Health Sciences, College of Medicine, National Cheng Kung University, Tainan 70101, Taiwan; juliehsinyun@gmail.com (H.-Y.C.); jerhao@mail.ncku.edu.tw (J.-H.C.); 2Department of Family Medicine, National Cheng Kung University Hospital, College of Medicine, National Cheng Kung University, Tainan 70403, Taiwan; yinfan@mail.ncku.edu.tw; 3Department of Family Medicine, Tainan Hospital, Ministry of Health and Welfare, Tainan 70043, Taiwan; 4Department of Occupational Therapy, College of Medicine, National Cheng Kung University, Tainan 70101, Taiwan; 5Department of Family Medicine, College of Medicine, National Cheng Kung University, Tainan 70101, Taiwan; 6Institute of Gerontology, College of Medicine, National Cheng Kung University, Tainan 70101, Taiwan

**Keywords:** metabolic health, obesity, medical personnel, hospital workers

## Abstract

The distribution of metabolic and obesity phenotypes in Taiwanese medical personnel is unknown. In this study, trajectory analysis with repeated measurements was used to explore the development and associated risk factors of different metabolic and obesity phenotypes in hospital staff from a Taiwanese medical center. The results demonstrated that metabolically unhealthy workers presented with a higher body mass index (BMI) compared with their metabolically healthy counterparts. Male and aged > 40 years hospital workers were more likely to be in a deleterious metabolic/obesity state. Meanwhile, profession and working hours were not significantly associated with the development of certain phenotypes in our study. These results shed light on the necessity of adequate data retrieval regarding working hours, and a nuanced examination of working conditions among different professions. Our findings are helpful for the development of advanced guidance regarding health promotion in hospital workers.

## 1. Introduction

Metabolic health and obesity have received attention across various occupations, especially in healthcare workers [1,2,3,4,5], and previous studies from England and the US revealed that hospital workers had a high prevalence of obesity [3,4]. Metabolic syndrome (MetS)often occurs in people who are overweight and obese, and it is defined as a cluster of at least three of the following five clinical risk factors: abdominal obesity, hypertension, elevated fasting sugar, elevated serum triglycerides and low serum high-density lipoprotein cholesterol (HDL-C) [6]. MetS and obesity are at the core of health promotion policies for hospital staff, both because of the shared pathophysiological mechanisms leading to subsequent cardiovascular complications, and also the modifiable and measurable nature of the diagnostic components [2,3]. While prior research has focused on obesity and MetS separately, there is a paucity of studies examining the presentation of different metabolic and obesity phenotypes in medical personnel.

It has been previously shown that both metabolically healthy obese individuals and metabolically unhealthy normal weight subjects, had an increased risk of cardiovascular disease compared with those who were metabolically healthy and of normal weight [7,8]. Whether particular risk factors play a role in the development of certain metabolic and obesity phenotypes is debatable, but early intervention should be encouraged as opposed to late disease treatment. From a workplace standpoint, working hours is an important measure as it is adjustable. A study by the World Health Organization (WHO) and International Labor Organization estimated that long working hours (≥55 h/week) cause the attributable burdens of ischemic heart disease and stroke [9]. In healthcare workers, long working hours drew attention principally because burnouts from extended work shifts and overtime are not only linked with depressive state, but also the risk of cardiovascular disease due to increased sympathetic tone after mental stress [10], causing a negative impact on patient safety [11,12]. Furthermore, occupational distribution based on metabolic and obesity phenotypes in this population also remain unknown.

This study aimed to use a trajectory analysis to identify the development of different metabolic and obesity phenotypes as well as associated factors in hospital staff. The results will help employers recognize risk factors regarding the health status of healthcare personnel and to adopt corresponding management measures. Findings from this study may be used for health management and policy development for hospital workers.

## 2. Materials and Methods

### 2.1. Data Sources and Study Population

This study was conducted in a Taiwanese medical center with employees divided into four groups: physicians, nurses, other medical staff (including pharmacist, dietitian and radiation technologist), and administrative staff. According to the Occupational Safety and Health Act in Taiwan, workers should receive regular health examinations and the frequency of these check-ups should be based on the worker’s profession, age, and seniority. Age cutoff of 40 years was applied for group stratification, as employees above 40 years would receive more extensive examinations than the younger workers. We retrospectively collected the health examination results from hospital employees conducted between 2012 and 2020. Employees with up to four repeated health examination results were enrolled for trajectory analysis. We excluded outsourcers, part-time workers, those aged ≤ 20 years or with missing data, as the long-term follow-up were not completed in these populations. The health examination results included body weight, height, serum biochemistry profile, a questionnaire with a self-reported medical history, health behavior characteristics, and working hours per week. The research protocol and access to the data were approved by the institutional review board of the National Cheng Kung University Hospital (IRB number: B-ER-110–008). The need for informed consent was waived because of the retrospective study design.

### 2.2. Measurements Metabolic Health, Overweight and Obesity

The metabolic and obesity phenotypes were defined based on a combination of the participants metabolic state and body mass index (BMI). In our study, metabolic health was defined according to the Adult Treatment Panel III and the modified American Diabetes Association criteria for impaired fasting glucose [13,14]. Being metabolically unhealthy was defined as presenting with any one of the following abnormalities: high blood pressure (systolic ≥ 130 mmHg/diastolic ≥ 85 mmHg, or a self-reported history of hypertension, or taking antihypertensive medications), hyperglycemia (fasting glucose ≥ 100 mg/dL, or taking medications for diabetes), hypertriglyceridemia (fasting triglycerides ≥ 150 mg/dL, or taking triglyceride-lowering medications), low high-density lipoprotein cholesterol (HDL-C < 40 mg/dL in men, <50 mg/dL in women, or a self-reported history of dyslipidemia, or taking lipid-controlling medication). Being metabolically healthy was defined as presenting with none of the metabolic disorders detailed above [15]. The subject’s weight and height were taken by trained staff, with weight measured to the nearest 0.1 kg and height measured to the nearest 0.1 cm, using a super-view automatic height measurement and weighing scale (HW-3030). BMI was calculated as weight in kilograms divided by height in meters squared, and was then divided into non-overweight (NO, BMI < 24 kg/m^2^) and overweight/obese (OO, BMI ≥ 24 kg/m^2^) [16] according to the criteria defined by the Department of Health in Taiwan.

#### 2.2.1. Metabolic and Obesity Phenotypes

According to their BMI and metabolic status groupings, all subjects were cross-classified into four phenotypes: metabolically healthy non-overweight (MHNO), metabolically unhealthy non-overweight (MUNO), metabolically healthy overweight/obese (MHOO), and metabolically unhealthy overweight/obese (MUOO). Due to the small numbers of underweight subjects, the underweight and normal weight categories were aggregated into a single category as non-overweight for analysis. The overweight and obese phenotypes were also combined given their relatively small sample sizes.

#### 2.2.2. Working Hours and Health Behavior Characteristics

For working hours, the question, “What are the average weekly working hours at work in the past 6 months?” was used, and the self-report results were further categorized into three groups: ≤40, 41–49, and ≥50 h/week. Fewer than 20 employees reported smoking, and therefore this parameter was not included in the analysis.

### 2.3. Statistical Analysis

Subjects with complete data from four health examinations were included in the longitudinal data analysis. Each subject could be classified into one of the four phenotypes for metabolic status at each wave. However, the phenotype of metabolic status for each subject could be different among different waves. For example, a subject could have been classified as MHNO at wave 1, MUNO at wave 2, MHOO at wave 3, and finally MUOO at wave 4, whereas another subject could have remained MHNO through all waves. Therefore, each subject had his/her own developmental trajectory of one of the metabolic and obesity states over time. The latent classes (grouping) of the developmental trajectory of each metabolic state over the four waves were specified using the group-based trajectory model (GBTM). The GBTM assumes that the population is composed of a mixture of latent classes (groups) with distinctive developmental trajectories. The Bayesian information criterion (BIC) index was used to select the optimal model among the models tested, which included a different number of latent classes and different shapes (linear, quadratic and cubic trends) [17]. In addition, for the chosen optimal model, the minimum sample size of a given trajectory was required to be ≥5% of the total population.

Since there were four metabolic and obesity phenotypes, the GBTM was conducted on each phenotype separately. After identifying the latent classes of trajectories for each metabolic and obesity phenotype, the association between basic demographics and the latent classes of trajectories for each state was studied using multivariable logistic regression analysis. A two-sided *p* value < 0.05 was considered to indicate a statistically significant difference. All statistical analyses were performed using SAS version 9.4 (SAS Institute, Cary, NC, USA), including the macro ‘PROC TRAJ’ for GBTM [18].

## 3. Results

### 3.1. Latent Classes of the Developmental Trajectories for the Metabolic and Obesity Phenotypes

A total of 1169 employees with four examinations between 2012 and 2020 were included in the GBTM. The results revealed that the number of latent classes (groups) with the best fit for the GBTMs for MHNO, MUNO, MHOO and MUOO were 3, 2, 2 and 3, respectively. The developmental trajectories of 1169 individuals for MHNO were divided into group 1 (size: 35.7%) persistently low probability (to be MHNO); group 2 (22.6%) initially moderate followed by decreasing probability (to be MHNO); and group 3 (41.8%) consistently high probability (Figure 1A). The 1169 individual trajectories for MUNO were divided into group 1 (80.1%) persistently low probability; and group 2 (19.9%) moderate and then slightly increasing probability (Figure 1B). The 1169 individual trajectories for MHOO were divided into group 1 (86.7%) persistently low probability; and group 2 (13.3%) initially increased followed by decreased probability (Figure 1C). The 1169 individual trajectories for MUOO were divided into group 1 (65.8%) persistently low probability; group 2 (14.1%) increasing probability; and group 3 (20.1%) persistently high probability (Figure 1D).

### 3.2. Factors Associated with the Latent Classes of Developmental Trajectories for Each Phenotype

The demographics among different latent classes (groups) of trajectories are summarized in Table 1, Table 2, Table 3 and Table 4. Using group 3 (persistently high probability) as the reference group in the analysis of latent classes for MHNO, age ≥ 40 years and male were associated with a higher likelihood of being in group 1 (persistently low probability) and group 2 (initially moderate followed by decreasing probability). In addition, subjects with alcohol consumption were significantly more likely to be in group 2 (OR 2.31, 95% CI 1.22–4.39). No significant association was identified between working hours or profession type and trajectory groups (Table 1).

Using group 1 (persistently low probability) as the reference group in the analysis of latent classes for MUNO, age ≥ 40 years and male were associated with a higher likelihood of being in group 2 (moderate and then slightly increasing probability) (Table 2).

Analysis of latent classes for MHOO revealed that no factors were associated with the different latent classes (Table 3). Using group 1 (persistently low probability) as the reference group in the analysis of latent classes for MUOO, male and those with alcohol consumption were associated with a higher likelihood of being in group 2 (increasing probability). The results also suggested that subjects aged ≥40 years and male were more likely to be in group 3 (persistently high probability) (Table 4).

In the analysis of MHNO, group 1 (persistently low probability of being MHNO) presented with a higher BMI from wave 1 through wave 4 compared with group 2 (initially moderate followed by decreasing probability) and group 3 (persistently high probability). In the analysis of MUOO, group 2 (increasing probability) and group 3 (persistently high probability) also presented with a significantly higher BMI compared with group 1 (persistently low probability of being MUOO) (Table 5). The BMI values at each wave with different latent classes of the trajectories for each metabolic state are provided in the Appendix A.

## 4. Discussion

Single time point on-site screening in the workplace often treats obesity and metabolic disorders as separate entities instead of integrating the two as different phenotypes. To the best of our knowledge, this is the first study to use metabolic/obesity phenotypes to investigate the health of hospital workers in Taiwan. In our study, hospital workers who were male, elder or reported alcohol consumption were prone to develop into metabolically unhealthy or overweight/obese state. In addition, longer working hours were not shown to be associated with a risk of worsened metabolic health or developing obesity.

Obesity and metabolic derangements often coexist and increase the risk of type 2 diabetes and cardiovascular disease [13,19]. Overweight and especially obese subjects are more likely to go through metabolic deterioration compared with normal-weight individuals [20]. However, there is not a unified criteria defining the metabolic/obesity phenotypes. BMI ≥ 25 kg/m^2^ and ≥30 kg/m^2^ are used by the National Institute of Health (NIH) as well as the WHO to define overweight and obese individuals, respectively [19]; however, this might underestimate the obesity risk in Asian populations. The WHO-Asian criteria define overweight as a BMI ≥ 23 kg/m^2^ and obesity as a BMI ≥ 25 kg/m^2^ [13]. The Ministry of Health and Welfare of Taiwan defines obesity as a BMI  ≥ 27 kg/m^2^ and overweight as a BMI of between 24 and 27 kg/m^2^ based on local statistical results [16,21]. The official annual report about obesity prevention in Taiwan also categorized subjects falling within the overweight and obese range (BMI ≥ 24 kg/m^2^) together [22]. For the above reasons, our study classified participants into non-obese and overweight/obese groups with an ethnicity-specific BMI cut-off point of 24 kg/m^2^ without further stratification into overweight and obese groups. Regarding metabolic health, numerous studies have attempted to create a standardized definition [23,24], with some involving the parameters of high sensitivity C-reactive protein and high homeostasis model assessment of insulin resistance [25]. In some studies, being metabolically healthy may be defined as having either 0, 1, or 2 metabolic syndrome components, and people reported as being metabolically healthy simply had fewer metabolic abnormalities than those with metabolic syndrome [26]. We defined metabolic health strictly as having no metabolic disorders or medication use for blood pressure, fasting sugar or lipids based on routine labor examinations and a self-report questionnaire. This classification was also consistent with the policy for recruiting hospital workers for health promotion in the study hospital.

A previous review suggested that although they were within the normal weight range, metabolically unhealthy normal weight subjects are significantly “more obese” than their metabolically healthy lean peers [27], and our results support this hypothesis. The MHNO group presented with the lowest BMI (20.7 ± 1.9 kg/m^2^) followed by the MHOO group with a BMI of 26.3 ± 2.3 kg/m^2^; both their metabolically unhealthy counterparts had a higher BMI (MUNO, BMI 21.3 ± 1.9 kg/m^2^; MUOO, BMI 28.1 ± 3.6 kg/m^2^) (Table A1). In the MUOO trajectory, all 3 groups presented with a trend of elevating BMI through wave 1 to wave 4 (Table 5). Group 2, which had an increasing probability of becoming MUOO over the years, showed even more prominent changes in the BMI than groups 1 and 3. Overall, BMI can be considered more than just cut-off points for weight category. The more obese subjects with a higher baseline and follow-up BMI may be associated with a higher risk of developing into deleterious metabolic/obesity phenotypes. Even in subjects within non-obese range, BMI may also serve as predictor of metabolic deterioration.

While a positive relationship between age and MetS has been reported [28,29], the relationship between sex and MetS has been inconsistent. Several studies revealed conflicting results of the sex differences in prevalence of MetS from disparate populations [5,30,31,32,33]. Whether the sex differences exist with certain distribution in metabolic/obesity phenotypes is unclear. Our study showed hospital workers with male sex dominance of developing into metabolically unhealthy or overweight/obese state despite of the smaller number compared to female employees, and the effect was pronounced in the group 2 (increasing probability) trajectory of MUOO (Table 4). Identification, targeting follow-up and intervention in such hospital workers may be warranted in future.

The occupational distribution of both obesity and metabolic syndrome have been shown in previous studies [34,35,36,37]. For healthcare workers, both issues have raised considerable attention over the years. Long working hours have been shown to lead to burnout [12], and to be associated with MetS and cardiovascular risk factors [38,39,40]. However, previous studies showed inconsistent results regarding working hours, metabolic health and obesity. A cross-sectional study conducted in a university hospital with predominantly female workers, revealed associations between metabolic syndrome and regular office hours shifts, and prolonged period of employment, but not with sex or night shifts [2]. Another cross-sectional study from Japan showed that longer working hours (>10 h/day) may contribute to an increased risk of metabolic syndrome in male workers [39]. A Korean study found associations between increased working hours and MetS in female but not male workers after adjustment for occupational characteristics [41]. The SUN project, a Spanish prospective cohort study, following participants for 8.3 years to explore working hours and the incidence of metabolic syndrome, suggested that long working hours did not increase the risk of developing MetS or each of its components [42]. In our study, duration of working hours was not significantly associated with worsened metabolic health or developing obesity in hospital workers. However, both the SUN project and our study took working hours only at a single time point and generated working hours data via a self-report questionnaire. According to the Labor Standards Act in Taiwan, the regular working time of workers may not exceed 8 h a day or 40 h a week. A self-report of working hours may not be entirely accurate as workers might avoid providing answers which contradict this well-known regulation. Recall bias might also cause a discrepancy between subjective working hours and objective working hours. This stands in line with a previous study which highlighted that subjective working hours should be interpreted carefully when assessing the health effects of long working hours [43]. In addition, as working hours were examined as a categorical variable, potential curvilinear effects on metabolic and obesity status might also have been neglected.

A prior study on MetS in Taiwanese hospital workers showed that physicians and administrative staff had a higher prevalence of MetS compared with other professions [5]. It argued that long working hours in physicians resulted in less sleep and exercise, and that the sedentary working style of administrative staff contributed to a higher prevalence of MetS. Conversely, physicians from our study presented with a lower risk of being MUOO compared with nurses and administrative staff in a cross-sectional analysis (Table A2); but this association was diminished in the trajectory analysis. This may be explained by the healthy worker effect exhibited in physicians [44]; the fact that lifestyle patterns vary greatly among physicians from different specialties, and because a smaller sample size were enrolled for trajectory analysis.

### Limitations

This was a single-institution study. There was a lack of information regarding working conditions and lifestyles, and some of the parameters that may have helped identify risk factors for overweight/obesity and being metabolically unhealthy in hospital workers remain unknown. Details of shift work were not investigated in the questionnaire until the latest amendments to the Occupational Safety and Health Act of Taiwan in December 2021. Since different departments allocate work between different professions uniquely, the proportion of sustained sedentary time, specific shifts that hospital workers are on, and whether an appropriate rotation system is adopted should be explored in relation to its potential effect on metabolic/obesity state.

## 5. Conclusions

In conclusion, metabolically unhealthy workers presented with a higher BMI compared with their metabolically healthy counterparts at a tertiary hospital in Taiwan. From a public health and preventive medicine perspective, early health interventions should be encouraged, as this may be more efficient than late treatment. Further investigations addressing how these phenotypes convert into different states or develop into significant cardiovascular diseases are warranted. Our findings did not show that longer working hours were associated with a risk of worsened metabolic health or developing obesity; however, future studies may generate data regarding shift work, and both subjective and objective working hours to explore their potential associations between different aspects of health in hospital workers. Details of shift patterns as well as working conditions among different professions, should also be inspected. Hopefully, our preliminary results can help raise awareness about the need for developing active guidance on monitoring the metabolic health and overweight/obesity status of Taiwanese hospital workers.

## Figures and Tables

**Figure 1 ijerph-19-08184-f001:**
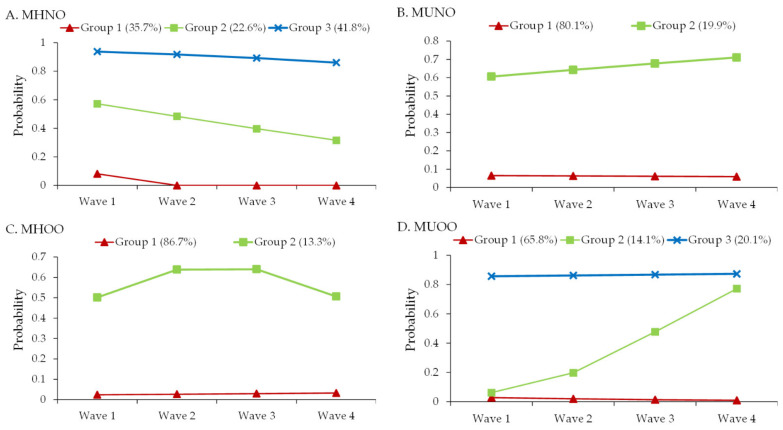
The best-fitting group-based trajectory models of (**A**) MHNO, (**B**) MUNO, (**C**) MHOO, and (**D**) MUOO. MHNO, metabolically healthy non-overweight; MUNO, metabolically unhealthy non-overweight; MHOO, metabolically healthy overweight/obesity; MUOO, metabolically unhealthy overweight/obesity.

**Table 1 ijerph-19-08184-t001:** Basic demographics of the study population according to the latent classes of developmental trajectories for MHNO in the longitudinal study.

	Descriptive Statistics			Logistic Model, Reference: Group 3			
	Group 1	Group 2	Group 3	Group 1		Group 2	
Variable	(*n* = 417)	(*n* = 264)	(*n* = 488)	Adjusted OR (95% CI)	*p*-Value	Adjusted OR (95% CI)	*p*-Value
Age (years)	38.2 ± 8.9	35.6 ± 9.2	33.0 ± 8.6				
Age (years)							
20–40	200 (48.0)	153 (58.0)	340 (69.7)	Reference		Reference	
≥40	217 (52.0)	111 (42.0)	148 (30.3)	2.39 (1.75–3.27)	<0.001	1.61 (1.15–2.27)	0.006
Male sex	122 (29.3)	41 (15.5)	20 (4.1)	12.24 (6.47–23.15)	<0.001	5.90 (2.92–11.94)	<0.001
Alcohol consumption	387 (92.8)	251 (95.1)	431 (88.3)	1.36 (0.81–2.30)	0.244	2.31 (1.22–4.39)	0.010
Working hours/week	48.4 ± 12.0	45.7 ± 8.1	45.6 ± 8.8				
Working hours/week							
≤40	161 (38.6)	107 (40.5)	212 (43.4)	Reference		Reference	
41–49	109 (26.1)	83 (31.4)	143 (29.3)	1.00 (0.71–1.42)	0.986	1.11 (0.77–1.60)	0.578
≥50	147 (35.3)	74 (28.0)	133 (27.3)	1.09 (0.76–1.55)	0.647	0.91 (0.61–1.36)	0.652
Profession type							
Nurse	202 (48.4)	166 (62.9)	338 (69.3)	Reference		Reference	
Physician	72 (17.3)	30 (11.4)	34 (7.0)	0.70 (0.37–1.35)	0.291	0.69 (0.34–1.41)	0.311
Other medical staff	54 (12.9)	30 (11.4)	41 (8.4)	1.20 (0.74–1.95)	0.460	1.03 (0.61–1.76)	0.905
Administrative staff	89 (21.3)	38 (14.4)	75 (15.4)	1.06 (0.71–1.57)	0.787	0.72 (0.46–1.14)	0.166

MHNO, metabolically healthy non-overweight; OR, odds ratio; CI, confidence interval; data were presented as frequency (percentage) or mean ± standard deviation; Group 1: Persistently low probability; Group 2: Initially moderate followed by decreasing probability; Group 3: Persistently high probability.

**Table 2 ijerph-19-08184-t002:** Basic demographics of the study population according to the latent classes of developmental trajectories for MUNO in the longitudinal study.

Variable	Descriptive Statistics		Logistic Model, Reference: Group 1	
	Group 1 (*n* = 936)	Group 2 (*n* = 233)	Adjusted OR (95% CI)	*p*-Value
Age (years)	34.7 ± 8.9	38.5 ± 9.4		
Age (years)				
20–40	588 (62.8)	105 (45.1)	Reference	
≥40	348 (37.2)	128 (54.9)	1.80 (1.31–2.47)	<0.001
Male sex	136 (14.5)	47 (20.2)	1.66 (1.02–2.68)	0.040
Alcohol consumption	845 (90.3)	224 (96.1)	2.04 (1.00–4.18)	0.051
Working hours/week	46.9 ± 10.4	45.7 ± 7.8		
Working hours/week				
≤40	378 (40.4)	102 (43.8)	Reference	
41–49	271 (29.0)	64 (27.5)	0.84 (0.59–1.21)	0.358
≥50	287 (30.7)	67 (28.8)	0.83 (0.57–1.21)	0.332
Profession type				
Nurse	584 (62.4)	122 (52.4)	Reference	
Physician	112 (12.0)	24 (10.3)	0.76 (0.40–1.43)	0.391
Other medical staff	89 (9.5)	36 (15.5)	1.38 (0.87–2.20)	0.171
Administrative staff	151 (16.1)	51 (21.9)	1.16 (0.78–1.74)	0.460

MUNO, metabolically unhealthy non-overweight; OR, odds ratio; CI, confidence interval; Data were presented as frequency (percentage) or mean ± standard deviation; Group 1: Persistently low probability; Group 2: Moderate and then slightly increasing probability.

**Table 3 ijerph-19-08184-t003:** Basic demographics of the study population according to the latent classes of developmental trajectories for MHOO in the longitudinal study.

Variable	Descriptive Statistics		Logistic Model, Reference: Group 1	
	Group 1 (*n* = 1014)	Group 2 (*n* = 155)	Adjusted OR (95% CI)	*p*-Value
Age (years)	35.6 ± 9.2	34.5 ± 8.5		
Age (years)				
20–40	592 (58.4)	101 (65.2)	Reference	
≥40	422 (41.6)	54 (34.8)	0.71 (0.48–1.04)	0.078
Male sex	159 (15.7)	24 (15.5)	1.06 (0.58–1.95)	0.850
Alcohol consumption	927 (91.4)	142 (91.6)	1.15 (0.62–2.15)	0.661
Working hours/weeks	46.4 ± 9.6	47.9 ± 11.9		
Working hours/weeks				
≤40	422 (41.6)	58(37.4)	Reference	
41–49	287 (28.3)	48 (31.0)	1.25 (0.82–1.89)	0.300
≥50	305 (30.1)	49 (31.6)	1.27 (0.82–1.97)	0.289
Profession type				
Nurse	610 (60.2)	96 (61.9)	Reference	
Physician	118 (11.6)	18 (11.6)	0.85 (0.41–1.77)	0.670
Other medical staff	106 (10.5)	19 (12.3)	1.25 (0.71–2.19)	0.441
Administrative staff	180 (17.8)	22 (14.2)	0.87 (0.52–1.46)	0.593

MHOO, metabolically healthy overweight/obesity; OR, odds ratio; CI, confidence interval; Data were presented as frequency (percentage) or mean ± standard deviation; Group 1: Persistently low probability; Group 2: Initially increased followed by decreased probability.

**Table 4 ijerph-19-08184-t004:** Basic demographics of the study population according to the latent classes of developmental trajectories for MUOO in the longitudinal study.

	Descriptive Statistics			Logistic Model, Reference: Group 1			
	Group 1	Group 2	Group 3	Group 2		Group 3	
Variable	(*n* = 769)	(*n* = 165)	(*n* = 235)	Adjusted OR (95% CI)	*p*-Value	Adjusted OR (95% CI)	*p*-Value
Age (years)	34.5 ± 9.1	35.2 ± 8.4	38.8 ± 8.9				
Age (years)							
20–40	480 (62.4)	107 (64.8)	106 (45.1)	Reference		Reference	
≥40	289 (37.6)	58 (35.2)	129 (54.9)	0.82 (0.56–1.21)	0.314	2.07 (1.48–2.89)	<0.001
Male sex	68 (8.8)	39 (23.6)	76 (32.3)	3.35 (1.89–5.93)	<0.001	5.42 (3.26–9.02)	<0.001
Alcohol consumption	696 (90.5)	155 (93.9)	218 (92.8)	1.74 (0.86–3.51)	0.122	1.14 (0.63–2.08)	0.663
Working hours/week	45.7 ± 8.5	46.9 ± 11.3	49.5 ± 12.5				
Working hours/week							
≤40	329 (42.8)	68 (41.2)	83 (35.3)	Reference		Reference	
41–49	224 (29.1)	52 (31.5)	59 (25.1)	1.17 (0.78–1.77)	0.440	1.07 (0.72–1.58)	0.741
≥50	216 (28.1)	45 (27.3)	93 (39.6)	0.92 (0.58–1.45)	0.723	1.30 (0.88–1.91)	0.188
Profession type							
Nurse	498 (64.8)	90 (54.5)	118 (50.2)	Reference		Reference	
Physician	64 (8.3)	25 (15.2)	47 (20.0)	0.97 (0.48–1.98)	0.943	0.83 (0.44–1.56)	0.561
Other medical staff	78 (10.1)	25 (15.2)	22 (9.4)	1.42 (0.83–2.45)	0.205	0.65 (0.36–1.14)	0.133
Administrative staff	129 (16.8)	25 (15.2)	48 (20.4)	0.90 (0.54–1.52)	0.697	0.87 (0.56–1.35)	0.544

MUOO, metabolically unhealthy overweight/obesity; OR, odds ratio; CI, confidence interval; Data were presented as frequency (percentage) or mean ± standard deviation; Group 1: Persistently low probability; Group 2: Increasing probability; Group 3: Persistently high probability.

**Table 5 ijerph-19-08184-t005:** The difference in BMI according to the latent classes of developmental trajectories for different metabolic and obesity phenotypes.

Metabolic and Obesity Phenotype/Latent Classes	BMI at Wave 1	BMI at Wave 4	Change (95% CI)	*p*-Value
MHNO				
Group 1	26.4 ± 3.9	27.3 ± 3.9	0.90 (0.68, 1.11)	<0.001
Group 2	21.9 ± 2.3	23.3 ± 2.5	1.39 (1.12, 1.66)	<0.001
Group 3	20.3 ± 2.0	21.1 ± 2.0	0.81 (0.68, 0.95)	<0.001
MUNO				
Group 1	23.2 ± 4.2	24.2 ± 4.3	1.01 (0.88, 1.14)	<0.001
Group 2	21.3 ± 1.9	22.1 ± 1.8	0.83 (0.63, 1.03)	<0.001
MHOO				
Group 1	22.4 ± 4.0	23.3 ± 3.9	0.90 (0.78, 1.02)	<0.001
Group 2	25.6 ± 2.5	27.1 ± 2.6	1.45 (1.09, 1.81)	<0.001
MUOO				
Group 1	21.0 ± 2.4	21.7 ± 2.4	0.78 (0.66, 0.89)	<0.001
Group 2	24.1 ± 2.4	26.6 ± 2.5	2.42 (2.11, 2.74)	<0.001
Group 3	28.1 ± 3.8	28.7 ± 3.7	0.60 (0.27, 0.94)	<0.001

## Data Availability

Due to the nature of this research, participants of this study did not agree for their data to be shared publicly, so Appendix A is not available.

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
