# Peer review of "Metabolic and Obesity Phenotype Trajectories in Taiwanese Medical Personnel"

_ijerph, 2022, doi:10.3390/ijerph19138184_

Round 1

Reviewer 1 Report

General remarks:

It is an interesting study by Chang et al. in which they have analyzed metabolic and obesity phenotype trajectory among healthcare workers in Taiwan. They have reported data of 1169 individuals that were followed for 8 years between 2012 and 2020 with 4 examinations during this period making it an important contribution to obesity and metabolic disorder literature. However, The manuscript needs rewriting and a better presentation to convey the data to the audience in an effective manner.

Specific comments:

  1. Please explain the methods of statistical analysis in more detail for a better understanding of the readers.
    1. Explain GTBM.
    2. What are these trajectories?
    3. Probability of which event has been plotted in the figures 1a-1d? Please explain in methods, mention in results, and also label the figure axis accordingly.
  2. Line 131: Please elaborate on the “low probability”, what probability is being measured here? Same in line 132 “decreasing probability”.
  3. Section 3.1. : What do these different trajectories mean? Please elaborate, since it is essential for the understanding of the paper.
  4. Line 166: What is “MUNO analysis”? It has not been explained in the methods.
  5. Line 178: what is “MUHO analysis”?
  6. A graph showing BMI change over time in MHNO, MUNO, MHOO, and MUOO groups would be an important and useful addition to the manuscript. Please add.
  7. All the groups (MHNO, MUNO, MHOO, and MUOO) have 1169 individuals, please explain. 
  8. Also in the “Metabolic and obesity phenotypes”(line 97) methods section, please mention the inclusion criteria for each category. 
  9. Line 211: Please include numbers to support the statement “the average BMI was higher in the metabolically healthy groups compared with their metabolically healthy counterparts”. Was this statistically significant? Also, this belongs to the results section.
  10. Line 217 to Line 240 should be shortened and moved to methods.
  11. Line 258: No data or table has been shown to support this statement. Please include a graph, and cite it along with this statement.
  12. Line 268-270: “A cross-sectional study conducted in a university hospital with predominantly female workers, revealed associations between metabolic syndrome and regular office hours shifts, and prolonged period of employment, but not with sex or night shifts” Unclear statement, please rephrase.

Author Response

1. Please explain the methods of statistical analysis in more detail for a better understanding of the readers.

    1. Explain GTBM.
    2. What are these trajectories?
    3. Probability of which event has been plotted in the figures 1a-1d? Please explain in methods, mention in results, and also label the figure axis accordingly.

Response 1: Thanks for your valuable comments. We have expanded the content of GBTM and explain the concept of “trajectories” more clearly in the “2.3. Statistical Analysis”. The probability denotes the likelihood to be in the latent class (group) of the specific metabolic state. We have made an effort to make it more clearly in the “2.3. Statistical Analysis” and “3. Results” including the title of the tables.

2. Line 131: Please elaborate on the “low probability”, what probability is being measured here? Same in line 132 “decreasing probability”.

Response 2: Thanks for your query. We have modified the sentences into “persistently low probability (to be MHNO)” and “decreasing probability (to be MHNO)”.

3. Section 3.1. : What do these different trajectories mean? Please elaborate, since it is essential for the understanding of the paper.

Response 3: Thanks for your comment. We have revised the text of Section 3.1 to make it clear to the readers.

4. Line 166: What is “MUNO analysis”? It has not been explained in the methods.

Response 4: Thanks for your query. We have rephrased it into “in the analysis of latent classes for MUNO”.

5. Line 178: what is “MUHO analysis”?

Response 5: Thanks for your query. We have rephrased it into “in the analysis of latent classes for MUHO”.

6. A graph showing BMI change over time in MHNO, MUNO, MHOO, and MUOO groups would be an important and useful addition to the manuscript. Please add.

Response 6: Thanks for your suggestion. We have provided the BMI value at each wave with different latent classes of the trajectories for each metabolic state in the newly-added Supplemental figure.

7. All the groups (MHNO, MUNO, MHOO, and MUOO) have 1169 individuals, please explain. 

Response 7: Thanks for your query. The GBTM was conducted on each metabolic state separately and therefore the sample size of the four metabolic states was same. We have explain it in the second paragraph in “2.3. Statistical Analysis”.

8. Also in the “Metabolic and obesity phenotypes”(line 97) methods section, please mention the inclusion criteria for each category. 

Response 8: The inclusion criteria of the whole trajectory analysis was mentioned in 2.1 Data Sources and Study Population: We retrospectively collected the health examination results from hospital employees conducted between 2012 and 2020. Employees with up to four repeated health examination results were enrolled for trajectory analysis. We excluded outsourcers, part-time workers, those aged ≤20 years or with missing data, as the long-term follow-up were not completed in these populations. The categorization of metabolic and obesity phenotypes is based on individual’s metabolic state (metabolically healthy vs. metabolically unhealthy) and BMI (non-overweight vs. overweight/obese), and there comes the four phenotypes in our study: MHNO, MUNO, MHOO, MUOO.

9. Line 211: Please include numbers to support the statement “the average BMI was higher in the metabolically healthy groups compared with their metabolically healthy counterparts”. Was this statistically significant? Also, this belongs to the results section.

Response 9: Thank you for the comment. Pertinent description is included from line 275 ~ 278 with cited table A1. The pre-existed redundant description is crossed out.

10. Line 217 to Line 240 should be shortened and moved to methods.

Response 10: Thank you for the suggestion. Since this paragraph elaborates how we define metabolic state, non-overweight and overweight/obesity due to intricated reasons and is tightly associated with interpretation of the results instead of simply showing the results, we prefer to keep it this way after giving great consideration.

11. Line 258: No data or table has been shown to support this statement. Please include a graph, and cite it along with this statement.

Response 11: Thanks for your query. The statement could be supported by table 1 (please see group 1—male employees were shown of adjusted OR 12.24 (p-value<0.001) to be in the “persistently low probability” to be MHNO) and table 4. We will cite table 4 specifically in this sentence.

12. Line 268-270: “A cross-sectional study conducted in a university hospital with predominantly female workers, revealed associations between metabolic syndrome and regular office hours shifts, and prolonged period of employment, but not with sex or night shifts” Unclear statement, please rephrase.

Response 12: Thanks for the suggestion. This statement serves as the beginning for the introduction of several published studies with inconsistent results regarding working hours, metabolic health and obesity. The writing aimed to show the widely varying outcomes in previous investigation, and was wrapped up with our interpretation and perspectives by the end of this paragraph.

Reviewer 2 Report

It´s relevant and interesting in research about Metabolic and Obesity Phenotype Trajectories in Taiwanese Medical Personnel, this topic is original and this paper give more details about the development of different metabolic and obesity phenotypes as well as associated factors in hospital staff.

Author Response

Thank you for the comments and appreciation.

Reviewer 3 Report

The Authors findings were helpful for the hospital workers in the development of advanced guidance regarding their health promotion. The tables are clear to understand the results. There are few typo and grammatical error that needs careful revision.

Author Response

Thank you for the comments and appreciation. We will proceed further revision for typo and grammatical errors.

Reviewer 4 Report

Dear Authors, I have read your manuscript with interest.

The current manuscript titled: "Metabolic and Obesity Phenotype Trajectories in Taiwanese Medical Personnel" represents an important analysis of evolving field of Internal Medicine and Public Health.

The title reflects the manuscript content and helps the reader navigate the article essence.

The abstract contains all the necessary information in a concise form.

In my opinion, these is the adjustment which should be made to increase the value of your manuscript:

1.      Line 20: add please abbreviation for BMI.

2.      Lines 20-23: this sentence is quite long, please, divide it into 2 parts.

3.     In the Introduction section, it is necessary to add a more detailed description of the metabolic syndrome and specify the obesity significance in this syndrome.

4.   Also, in the Introduction section, it is necessary to describe the effect of burnout syndrome on the cardiovascular system from a physiopathological point of view.

5.    Materials and Methods: Please explain why the administrative staff was included in this study. This structure does not have direct contact with patients, does not do duty/on-calls, etc. Please indicate your arguments and is this subgroup comparable with other subgroups?

6.      Line 87: please, add abbreviation for “HDL-C”.

7.   Line 94: in most BMI classifications, normal values are <25 kg/m2. In Discussions chapter, you pointed out that Taiwan uses a different categories, however, if this study is published for a wide range of readers from around the world, it is necessary to adapt the classifications used to internationally used ones.

8.   In Figure 1, please change the font so that it is the same as the main text of the manuscript.

9.      Line 152: in the Methods and Methods chapter, please indicate the age distribution. Moreover, please indicate why you used this distribution by age intervals.

10.   In Tables, change please “Administration” to “Administrative staff”.

11. In the study, you included data on the alcohol consumption by the studied candidates, however, specify if smoking was studied, as this parameter is of great importance in the metabolic syndrome and cardiovascular pathology. Try to add this data to the results.

12. In the Conclusions chapter, elements of methodology and results are repeated. Please try to make a brief synthesis of the study and indicate its main strengths and practical implications.

13. The manuscript contains some punctuation errors, please revise the text (lines 85, 86, 87, etc.).

Author Response

1. Line 20: add please abbreviation for BMI.

Response 1: Thanks for the query. Abbreviation is added.

2. Lines 20-23: this sentence is quite long, please, divide it into 2 parts.

Response 2: Thanks for the suggestion. The sentence is divided into 2 parts.

3. In the Introduction section, it is necessary to add a more detailed description of the metabolic syndrome and specify the obesity significance in this syndrome.

Response 3: Thanks for the query. We have rephrased the description and aimed to specify the importance of the idea of metabolic and obesity phenotypes.

4. Also, in the Introduction section, it is necessary to describe the effect of burnout syndrome on the cardiovascular system from a physiopathological point of view.

Response 4: Thank you for the suggestion. We have added relevant description. Though burnouts would lead to deteriorated cardiovascular outcomes, both burnouts and cardiovascular complications are manifestations of long working hours. Burnouts in this statement was mentioned aiming to present as adverse result of long working hours and to bring out the idea of negative impact on patient safety, instead of standing as contributing cause of poor cardiovascular outcomes.

5. Materials and Methods: Please explain why the administrative staff was included in this study. This structure does not have direct contact with patients, does not do duty/on-calls, etc. Please indicate your arguments and is this subgroup comparable with other subgroups?

Response 5: Thank you for the query. Administrative staff in Taiwanese medical personnel, especially in a tertiary medical center like our investigated hospital, does take on-calls/shifts, and share tremendous duty in patient care, including admission, discharge, emergency room patients disposition. The references we intend to compare (Yeh et al. 2018) have also encompassed administrative staff, as this population is unneglectable in our working environment.

6. Line 87: please, add abbreviation for “HDL-C”.

Response 6: Thank you. Abbreviation is added.

7. Line 94: in most BMI classifications, normal values are <25 kg/m2. In Discussions chapter, you pointed out that Taiwan uses a different categories, however, if this study is published for a wide range of readers from around the world, it is necessary to adapt the classifications used to internationally used ones.

Response 7: Thank you for the suggestion. Whether BMI and waist circumference cut points should be ethnicity specific have touched off international debate. The cut points for overweight and obesity being lower for Asians than for other ethnic groups has been advocated, though WHO has not agreed on setting different cutoff points for Asians given a lack of consensus among researchers as to what these lowered cutoffs should be. Not only Taiwan, but also China and Japan have proposed to define overweight as a BMI of 24. Meanwhile, International Diabetes Federation has included ethnic-specific criteria for the definition of abdominal obesity. We believe there is a need for the emergence of such research to support the argument of ethnic differences in BMI and disease risk, and our research would take this responsibility as well.

8. In Figure 1, please change the font so that it is the same as the main text of the manuscript.

Response 8: Thanks for the query. The font is changed.

9. Line 152: in the Methods and Methods chapter, please indicate the age distribution. Moreover, please indicate why you used this distribution by age intervals.

Response 9: Thank you for the query. We have added pertinent statement.

10. In Tables, change please “Administration” to “Administrative staff”.

Response 10: Thanks for the query. We have made the changes.

11. In the study, you included data on the alcohol consumption by the studied candidates, however, specify if smoking was studied, as this parameter is of great importance in the metabolic syndrome and cardiovascular pathology. Try to add this data to the results.

Response 11: Thank you for the query. Since less than 20 employees from the investigated hospital reported smoking, this parameter was not studied. We have added this statement as suggested.

12. In the Conclusions chapter, elements of methodology and results are repeated. Please try to make a brief synthesis of the study and indicate its main strengths and practical implications.

Response 12: Thank you for the important suggestion. We have modified the statement and still emphasized the importance of future perspectives and anticipation of further studies on this topic.

13. The manuscript contains some punctuation errors, please revise the text (lines 85, 86, 87, etc.).

Response 13: Thanks for the suggestion. We have revised the text accordingly.

Round 2

Reviewer 1 Report

The manuscript has improved significantly, I am pleased to recommend the acceptance of the manuscript for publication. 

However, there are some minor errors that need corrections.

Line 244: “In our study” is misspelled as “in out sturdy”, please correct this.

Line 253 and line 329: please cite relevant articles.

Author Response

1. Line 244: “In our study” is misspelled as “in out sturdy”, please correct this.

Response 1: Thank you for pointing this out. We have corrected the errors. 

2. Line 253 and line 329: please cite relevant articles.

Response 2: Thank you for the advice. We cited relevant articles.

Thank you for the valuable comments and feedback.

Reviewer 4 Report

The changes made have significantly increased the manuscript quality.

I recommend this article for publication.

Good luck!

Author Response

Thanks for all the valuable comments and feedback. Each of your insights have served to strengthen our manuscript. Thank you very much for the final recommendation.